# Glucocorticoid Receptor (GR) and Specificity Protein 1 (Sp1) or Sp3 Transactivate the Bovine Alphaherpesvirus 1 (BoHV-1)-Infected Cell Protein 0 Early Promoter

**DOI:** 10.3390/v17020229

**Published:** 2025-02-07

**Authors:** Sankha Hewawasam, Fouad S. El-Mayet, Clinton Jones

**Affiliations:** 1Department of Veterinary Pathobiology, College of Veterinary Medicine, Oklahoma State University, Stillwater, OK 74074, USA; shewawa@okstate.edu (S.H.); fouad.elmayet@okstate.edu (F.S.E.-M.); 2Department of Virology, Faculty of Veterinary Medicine, Benha University, Benha 13511, Egypt

**Keywords:** bovine alphaherpesvirus 1 (BoHV-1), Krüppel-like factor 4 (KLF4), glucocorticoid receptor (GR), Specificity protein 1 (Sp1), Specificity protein 3 (Sp3)

## Abstract

Bovine alphaherpesvirus 1 (BoHV-1) acute infection leads to latently infected sensory neurons in trigeminal ganglia. During lytic infection, the immediate early expression of infected cell protein 0 (bICP0) and bICP4 is regulated by an immediate early transcription unit 1 (IEtu1) promoter. A separate bICP0 early (E) promoter drives bICP0 as an early viral gene, presumably to sustain high levels during productive infection. Notably, bICP0 protein expression is detected before bICP4 during reactivation from latency, suggesting the bICP0 E promoter drives bICP0 protein expression during the early phases of reactivation from latency. The glucocorticoid receptor (GR) and Krüppel-like factor 4 (KLF4) cooperatively transactivate the bICP0 E promoter despite this promoter lacks a consensus GR response element (GRE). KLF and specificity protein (Sp) family members comprise a “super-family” of transcription factors. Consequently, we hypothesized Sp1 and Sp3 transactivated the bICP0 E promoter. These studies revealed GR and Sp3 or Sp1 cooperatively transactivated bICP0 E promoter activity. KLF4 and Sp3, but not Sp1, had an additive effect on bICP0 E promoter activity. Mutating the consensus Sp1 and CACCC binding sites proximal to the TATA box impaired promoter activity more than the Sp1 sites further upstream from the TATA box.

## 1. Introduction

Bovine alphaherpesvirus 1 (BoHV-1) infection impairs host immune responses but increases the incidence of life-threatening bacterial pneumonia [1,2]. Furthermore, BoHV-1 infection erodes mucosal surfaces and frequently causes conjunctivitis [3,4]. A BoHV-1 entry protein encoded by the poliovirus receptor-related 1 gene is an important bovine respiratory disease (BRD) susceptibility gene for Holstein calves [5]. Hence, BoHV-1 is a cofactor of BRD, a poly-microbial disease of cattle [4]. BRD is the most economically important disease that affects beef and dairy cattle because ~75% of morbidities and over 50% of all mortalities in feedlot cattle are linked to BRD [6,7,8,9]. *Mannheimia haemolytica (MH)* is a Gram-negative bacterium [10] that is part of the normal flora in the upper respiratory tract of healthy ruminants [11]. This commensal relationship is disrupted by stress and/or co-infections [2], which may lead to bacterial pneumonia. BoHV-1 can also induce abortions in pregnant cows following the infection of ovaries and/or the fetus [12].

Acute BoHV-1 infection induces apoptosis, inflammation, and high levels of virus production [13]. Viral gene expression occurs in three distinct phases as follows: immediate early (IE), early (E), or late (L). IE transcription unit 1 (IEtu1) encodes two transcriptional regulatory proteins (bICP0 and bICP4), which stimulate productive infection [14,15,16]. An early promoter also drives bICP0 expression [16] (Figure 1A), which we predict provides sufficient bICP0 protein levels during productive infection. IEtu2 encodes the viral protein bICP22 [15]. The tegument protein, bTIF, also referred to as VP16, specifically activates IE transcription by interacting with the TAATGARAT motif present in all IE promoters [17,18]. E proteins are non-structural and promote viral DNA replication. L proteins generally comprise the infectious virus particle.

Sensory neurons in trigeminal ganglia (TG) are a primary site for life-long latency following acute infection of oral, nasal, or ocular cavity [19,20]. Stress, food and water deprivation during the shipping of cattle, weaning, and/or dramatic weather changes increase corticosteroid levels and reactivation from latency [20,21]. The synthetic corticosteroid dexamethasone (DEX) consistently induces BoHV-1 reactivation in latently infected rabbits and calves [22,23]. Corticosteroids, including DEX, bind and activate the glucocorticoid receptor (GR) [24] suggesting GR regulates certain aspects of reactivation from latency. The Etu1 promoter, which drives immediate early bICP0 and bICP4 expression, contains two consensus GR response elements (GREs) in the promoter; mutating these GREs interferes with DEX-induced promoter activity [25]. Three IE genes (bICP0, bICP4, and bICP22) and VP16 are detected in TG neurons within 3 h of DEX treatment [26,27,28]. Interestingly, bICP0 is detected in TG neurons prior to bICP4, suggesting the bICP0 E promoter is activated sooner than the IEtu1 promoter. Conversely, other late viral proteins (gC and gE) are difficult to detect during reactivation. The immunosuppressive properties of increased corticosteroid levels are also predicted to enhance virus spread during the later stages of reactivation from latency [29].

Transcriptomic studies revealed cellular transcription factors are induced in TG neurons within 1–3 h after a single intravenous DEX injection is given to latently infected calves [30]. Strikingly, four members of the Krüppel-like factor (KLF) family were identified in this study—KLF4, KLF6, KLF15, and promyelocytic leukemia zinc finger (PLZF). KLF family members belong to the specificity protein (Sp) transcription factor family. This super-family of transcription factors interacts with GC- or CA-rich motifs [31,32]. This is relevant because the GC-rich BoHV-1 genome contains many Sp1 binding sites and Sp1 or Sp3 stimulate the BoHV-1 IEtu1 promoter [33] and HSV IE promoters [34,35]. Two pioneer transcription factors, GR and KLF4, synergistically transactivate the bICP0 E promoter [36]. Thus, GR and specific stress-induced transcription factors stimulate key viral promoters and productive infection following stress, as reviewed in [37].

The objectives of this study were to test whether Sp1 or Sp3 cooperate with GR to transactivate the bICP0 E promoter and stimulate productive infection. Furthermore, this study tested whether GR+KLF4-mediated transactivation was influenced by Sp1 or Sp3. Finally, the effect GR and Sp3 have on productive infection was examined.

**Figure 1 viruses-17-00229-f001:**
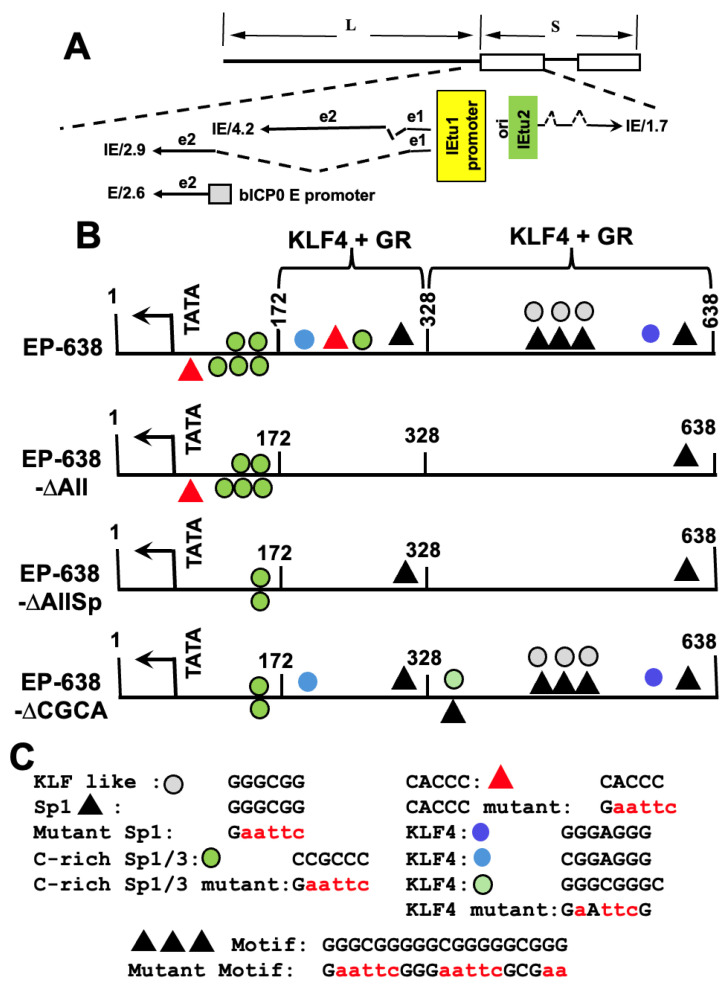
Schematic of BoHV-1 genome and immediate early transcription unit 1 (IEtu1). (**A**) BoHV-1 genome diagram. Sequences spanning unique long sequences are denoted as (L), unique short sequences are denoted by S, and the two open rectangles denote the repeats. The alternatively spliced IE transcripts (IE/4.2 and IE/2.9) are translated into bICP4 and bICP0, respectively. The EItu1 promoter (shaded in yellow) drives IE/4.2 and IE/2.9 expression during productive infection [15,16,38]. The expression of a second bICP0 transcript (E/2.6) is driven by the bICP0 early promoter (bICP0 E), which is shaded in gray. A distinct IE promoter, which is shaded in green, drives the expression of the IE/1.7 transcript, which is translated into the bICP22 protein. Solid lines in the transcripts denote exons (e1, e2, or e3) and dashed lines are introns. ORI denotes the origin of replication. (**B**) EP-638 was previously described [36,39,40] and cloned upstream of the luciferase vector (pGL3-Basic Vector, Promega (Madison, WI, USA)) as a SacI-HindIII fragment. The position of the TATA box is shown, and the arrow denotes the position and direction of the mRNA start site. The 5′ end of transcription is upstream of the initiating ATG. Two distinct regions in EP-638 are synergistically transactivated by KLF4 and GR [36,40]. Mutations in the respective EP-638 were synthesized by GenScript and fragments were inserted into the multiple cloning site of pGL3-Basic Vector as a SacI-HindIII fragment. (**C**) Summary of key transcription factor binding sites. Variants of KLF4 binding sites [41] are denoted by different colors of circles. These variants are influenced by their adjacent transcription factor binding sites and chromatin status. Mutations of the denoted Sp1 or KLF4 binding sites were disrupted by replacing the sequence with an EcoRI restriction site (GAATTCC).

## 2. Materials and Methods

### 2.1. Cells and Virus

Mouse neuroblastoma (Neuro-2A) cells were obtained from ATCC. These cells were grown in minimal essential medium (MEM) supplemented with 10% FBS, penicillin (10 U/mL), and streptomycin (100 µg/mL).

A BoHV-1 mutant containing the β-Gal gene in place of the viral gC gene was obtained from S. Chowdury (LSU School of Veterinary Medicine) (gCblue virus). This virus grows to similar titers as the wild-type (wt) parent virus and expresses the Lac Z gene. The preparation of BHV-1 genomic DNA was previously described [42,43,44,45]. Neuro-2A cells were co-transfected with the gCblue viral DNA using Lipofectamine 3000 (Invitrogen, Waltham, MA, USA). β-Galactosidase (β-Gal)-positive cells were counted at 24 h after transfection, as described previously [42,43,44,45]. The number of β-Gal+ cells in cultures transfected with gCblue genomic DNA was set at a value of 1 for each experiment and the effect of DEX and the expression of the mouse GR, KLF4, and/or Sp3 or expressed as a fold induction relative to the control. This representation of the data minimizes differences in cell density, variation in Lipofectamine 2000 lots, and transfection efficiency.

### 2.2. Plasmids

A mouse GR-α expression vector was obtained from Dr. Joseph Cidlowski, NIH, and is denoted as GR throughout the manuscript. The KLF4 expression vector was obtained from Dr. Jonathan Katz (University of Pennsylvania). The Sp1 (plasmid #27,264) and Sp3 (plasmid #24,541) expression vectors were purchased from Addgene.

The construction of the BoHV-1 bICP0 E promoter and deletion constructs (EP-638) used in the present study were described previously [46,47]. Mutants denoted in Figure 1 were synthesized by GenScript (Piscataway, NJ, USA). These bICP0 E promoter constructs are in the pGL3-Basic Vector (Promega, Madison, WI, USA; see Figure 1 for a schematic of these constructs). All plasmids were prepared from bacterial cultures by alkaline lysis and 2 rounds of cesium chloride ultracentrifugation.

### 2.3. Transfection and Dual-Luciferase Reporter Assay

Neuro-2A cells (8 × 10^5^) were seeded into 60 mm dishes containing MEM with 10% FBS at 24 h prior to transfection. Two hours before transfection, the medium was replaced with fresh growth medium lacking any antibiotics. Cells were co-transfected with the designated bICP0 E promoter construct which was cloned upstream of the firefly luciferase gene (0.5 µg plasmid DNA) in the pGL3-Basic Vector and a plasmid encoding *Renilla* luciferase under the control of a minimal herpesvirus thymidine kinase (TK) promoter (50 ng DNA) as a transfection control. Where indicated, the expression plasmids expressing GR, Sp1, or Sp3 (1 µg plasmid DNA) and/or KLF4 (0.5 µg plasmid DNA) were included in the plasmid mixture. To maintain equal plasmid concentrations in the transfection mixtures, an empty expression vector (pcDNA3.1) was added as needed. Transfections were performed with Lipofectamine 3000 (ThermoFisher cat# L3000015, Waltham, MA, USA), following the manufacturer’s guidelines. Neuro-2A cells were incubated in 2% charcoal-stripped FBS after transfection. At 24 h after transfection, the Neuro-2A cultures were treated with water-soluble DEX (10 µM; Sigma, D2915, St. Louis, MO, USA). Forty-eight hours after transfection, cells were harvested, and protein extracts were subjected to a dual-luciferase assay using a commercially available kit (E1910; Promega). Luminescence was measured by using a GloMax 20/20 luminometer (E5331; Promega).

### 2.4. Quantification of β-Gal-Positive Cells

Neuro-2A cells grown in 60 mm plates were co-transfected with 1.0 μg BoHV-1 gCblue DNA and, where indicated, a plasmid expressing mouse GR protein (1.0 μg), Sp3 (1.0 μg), and/or KLF4 (0.5 μg) using Lipofectamine 3000 (catalog no. L3000075; Invitrogen). To maintain the same amount of DNA in each sample, an empty vector was included in the samples. After transfection, 2% stripped FBS was added to the medium. At 48 h after transfection, cells were fixed with a solution containing 2% formaldehyde and 0.2% glutaraldehyde in phosphate-buffered saline (PBS) and then stained with a solution containing 1% Bluo-Gal, 5 mM potassium ferricyanide, 5 mM potassium ferrocyanide, and 0.5 M MgCl_2_ in PBS. The number of  β-Gal-positive cells was determined as described previously [47]. The number of β-Gal-positive cells in cultures expressing the blank vector was set at 1 for each experiment. To calculate the fold change of β-Gal-positive cells, the number of blue cells in cultures transfected with the plasmids of interest were divided by the number of blue cells in cultures transfected with the blank vector. The effect that the overexpression of KLF14, Sp3, and/or GR had on productive infection is expressed as fold induction relative to that of the control. This representation of the data minimizes differences in cell density, Lipofectamine 3000 lot variation, and transfection efficiency [47].

## 3. Results

### 3.1. GR and Sp1 Cooperatively Transactivate the bICP0 E Promoter

As mentioned above, IE expression of BoHV-1 bICP0 and bICP4 is regulated by the IEtu1 promoter (Figure 1A) [25,47,48]. Alternative splicing of the IEtu1 mRNA leads to two distinct mRNAs that are translated into bICP0 and bICP4. A distinct E promoter drives bICP0 expression with early kinetics [16]. The bICP0 E promoter is synergistically transactivated by two pioneer transcription factors GR and KLF4, which can specifically bind silent chromatin [36,40,41]. Since the mutagenesis of consensus Sp1 sites within the bICP0 E promoter significantly impaired GR- and KLF4-mediated transactivation, we hypothesized Sp1 family members transactivate the bICP0 E promoter. For these studies, we utilized a bICP0 E promoter construct denoted as EP-638, which contains several potential Sp1 and KLF binding sites (Figure 1B).

Initial studies tested whether Sp1 or GR and Sp1 increased EP-638 promoter activity. A previous study tested whether upstream GC-rich Sp1 binding sites and KLF4 binding sites were important for GR- and KLF4-mediated transactivation [36]. However, no mutations were made in the sequences between 1 and 172 of EP-628. Neuro-2A cells were used for these studies because they have neuronal-like properties [49] and are readily transfected. As previously demonstrated [36,40], GR transactivated EP-638 by approximately 8-fold, but the DEX addition reduced promoter activity (Figure 2). Regardless of DEX treatment, Sp1 stimulated EP-638 only two-fold; however, this was significantly higher than the wt EP-638 transfected with just an empty vector. When GR and Sp1 were co-transfected with EP-638, promoter activity was significantly higher than the EP-638 construct alone or when transfected with Sp1. Notably, EP-638 promoter activity in the presence of GR and Sp1, but not in the DEX treatment, exhibited a higher promoter activity than an additive effect by GR- or Sp1-mediated transactivation. However, there was no significant difference in promoter activity by GR and Sp1 when compared to the transactivation by GR, indicating that there were no cooperative effects on promoter activation.

EP-638 contains numerous GC-rich and CG-rich Sp1 consensus sequences in wt sequences (Figure 1B). There are also 5 KLF4 or KLF4-like motifs in EP-638, and 2 CACCC motifs. EP638-∆All retained these motifs located in the 172 bp region of EP-638. EP638-∆All had significantly lower promoter activity when co-transfected by GR and GR plus Sp1 (Figure 2). In contrast to EP638-∆All, EP638-∆AllSp1 contains a CACCC mutation adjacent to the TATA box and lacks 3 C-rich Sp1/3 sites but contained the G-rich Sp1 binding site near position 328. EP638-∆AllSp was transactivated by GR and Sp1 significantly less than EP638-∆All, suggesting that the C-rich Sp1 binding sites and/or the CACCC motif near the TATA box were important.

Interestingly, the Sp1-mediated transactivation of EP638-∆All and EP638-∆AllSp constructs was the same as that of the EP-638 wt construct (Figure 2). However, the GR-mediated transactivation of EP638-∆All and EP638-∆AllSp was significantly lower than that of the wt EP-638 construct. Relative to EP638-∆All or EP-638-∆AllSp, EP638-∆CGCA contains a KLF motif near position 172, a C-rich plus G-rich Sp1/3 motifs near position 328, and the triple KLF and G-rich Sp1 binding sites located between position 328 and 638. EP638-∆CGCA promoter activity was significantly increased when the construct was transactivated by GR and cultures treated with DEX when compared to the basal promoter activity. However, EP638-∆CGCA promoter activity was not significantly different than the empty vector, regardless of whether DEX was added to the cultures when transfected with Sp1. The GR- and Sp1-mediated transactivation of EP638-∆CGCA was significantly higher than basal promoter activity when cultures were not treated with DEX but was significantly lower than wt EP-638, EP638-All, and EP638-∆All. The EP638-∆CGCA construct contains mutations at the CG-rich Sp1 binding sites and a CACCC motif adjacent to the TATA box, indicating these motifs were more important than the Sp1 and KLF4 binding sites further upstream from the bICP0 E promoter TATA box.

### 3.2. Sp1 Does Not Cooperate with KLF4 to Transactivate the bICP0 E Promoter

Since GR and KLF4 synergistically transactivate the bICP0 E promoter [40], we tested whether KLF4 and Sp1 compete or cooperate to transactivate this important promoter. As expected, KLF4 transactivated EP-638 WT promoter activity approximately 30-fold, but the DEX treatment did not significantly increase promoter activity in Neuro-2A cells (Figure 3). Sp1 stimulated EP-638 WT promoter activity approximately 2-fold, which was consistent with the studies shown in Figure 2. Since KLF4 activated EP-638 more than 25-fold, the effect of Sp1 was not significantly different. As expected, the KLF4-mediated transactivation of EP638-∆All was significantly reduced when compared to the wt EP-638 construct. The KLF4-mediated transactivation of EP638-∆All was significantly higher than the EP638-∆AllSp1 construct. This result was intriguing because the differences between these two constructs is that the CACC-rich motif adjacent to the TATA box and the three C-rich binding sites are not present in the EP638-∆AllSp1 construct but the EP638-∆All construct contains these motifs. Consistent with the results obtained with GR and Sp1, the EP638-∆CGCA mutant exhibited the lowest promoter activity when co-transfected with KLF4 and Sp1. In summary, mutating the C-rich Sp1 sites and CACCC motif near the TATA box were more important than the upstream Sp1 and KLF4 binding sites.

### 3.3. GR and Sp3 Cooperatively Transactivate the bICP0 E Promoter

Consistent with previous studies, GR alone transactivated the EP-638 promoter approximately 7-fold, but the DEX addition slightly reduced promoter activity (Figure 4). Sp3, regardless of DEX treatment, stimulated the EP-638 promoter ~3-fold, which was significantly higher when compared to the promoter transfected with an empty vector. EP-638 promoter activity in the presence of both GR and Sp3, but without DEX treatment, was significantly higher than the additive effects of either GR or Sp3, indicating GR and Sp3 cooperatively transactivated EP-638 promoter activity. The addition of DEX did not significantly reduce transactivation by GR and Sp3. EP638-∆All exhibited significantly lower promoter activity when co-transfected with GR or Sp3. Although GR and Sp3 cooperatively transactivated EP638-∆All and EP638-∆AllSp relative to either GR or Sp3, promoter activity was significantly lower than the GR- and Sp3-mediated transactivation of wt EP-638. Furthermore, the GR-mediated transactivation of the EP638-∆AllSp construct was significantly lower than EP-638 and EP638-∆All.

The EP638-∆CGCA construct exhibited low levels of transactivation by GR or Sp3 and was not significantly different than when the construct was transfected with the empty vector. The GR- and Sp3-mediated transactivation of EP638-∆CGCA was significantly higher but significantly lower than EP-638, EP638-Sp, and EP638-∆AllSp. Although this result was unexpected because the EP638-∆CGCA construct contains the same mutations in the first 172 nucleotides as EP638-∆AllSp, these results underscore the complexity of the cooperative transactivation of the bICP0 E promoter by GR and Sp3 and suggest more than one cis-acting regulatory motif is important.

### 3.4. Sp3 and KLF4 Cooperatively Transactivated the bICP0 E Promoter

As expected, KLF4 transactivated EP-638 WT promoter activity approximately 30-fold and DEX treatment did not significantly increase promoter activity (Figure 5). Sp3 alone only transactivated the EP-638 wt promoter ~3-fold. Co-transfection with Sp3 and KLF4 resulted in an approximately 35-fold increased transactivation, which was significantly higher than just KLF4. The transactivation of EP638-∆All by KLF4 was greater than 20-fold, whereas the EP638-∆AllSp1 construct was not significantly transactivated by KLF. In contrast, the KLF4- and Sp3-mediated transactivation of EP638-∆All and EP638-∆All was greater than 10-fold but there was no significant difference between these two constructs. Although the EP638-∆CGCA mutant was transactivated by KLF4 and Sp3 approximately 10-fold, it was significantly higher than KLF4 or Sp3 alone. These studies suggested that mutating the C-rich Sp1 sites and CACCC motif close to the TATA box blocked KLF4-mediated transactivation of the bICP0 E promoter. These studies also suggest GC-rich Sp1 binding sites that are further upstream of the TATA box allowed for low levels of KLF4-mediated transactivation.

### 3.5. Sp3 Effects on GR- or KLF4-Mediated Transactivation Are Dose-Dependent

Our data demonstrated that GR and Sp3 cooperatively transactivate the EP-638 promoter using the same concentration of the respective transcription factors. Additional studies tested whether increasing levels of Sp3 increased or decreased promoter activity. Increasing Sp3 levels significantly reduced EP-638 promoter activity when co-transfected with GR (Figure 6A) and KLF4 (Figure 6A). These results confirmed that an equal concentration of Sp3 and GR or KLF4 was important for the efficient transactivation of the EP-638 promoter construct, whereas higher Sp3 concentrations reduced transactivation. Furthermore, increasing Sp3 levels reduced the KLF4-mediated transactivation of EP-638 (Figure 6B). Collectively, these studies revealed that a high concentration of Sp3 impaired GR- or KLF4-mediated transactivation. This study also suggests that Sp3 competes for KLF4 and GR binding to bICP0-E promoter sequences.

### 3.6. Sp3 Cooperates with GR and KLF4 to Stimulate Productive Infection

A previous study demonstrated KLF4 cooperates with GR and DEX to stimulate BoHV-1 productive infection [40]. Additional studies revealed GR and Sp1 or Sp3 transactivated the HSV-1 ICP0 promoter in Neuro-2A and CV-1 cells [35]. Finaly, a Sp1-specific siRNA significantly reduced BoHV-1 replication [33]. Hence, we focused on the effects Sp3, GR, and KLF4 had on BoHV-1 replication.

To examine the effect Sp3 and KLF4 have on the productive infection of BoHV-1, Neuro-2A cells were co-transfected with gCblue genomic DNA. We used the gCblue BoHV-1 recombinant, which contains the Lac Z gene downstream of the gC promoter, because this virus grows to similar titers as wt BHV-1 in bovine cells and the number of β-galactosidase positive cells correlate with plaque formation [45,50,51]. This approach was used instead of infecting cells and testing the effect of DEX on productive infection because we were concerned that VP16, which is part of the viral particle, would over-ride any effect DEX, viral, or cellular genes have on stimulating viral replication in permissive cells [30,45,46,47]. Neuro-2A cells were used for this study because they have neuronal-like properties [49] and are semi-permissive for BoHV-1 [52]. Twenty-four hours after transfection was the time-point used because it minimized the fusion of individual β-galactosidase-positive cells and secondary spread. GR and Sp3 plus DEX stimulated the number of β-galactosidase (β-Gal)-positive Neuro-2A cells more than 4-fold, which was significantly higher than that stimulated by GR plus DEX or Sp3 alone (Figure 7A,B). The co-transfection of gCblue, GR, and KLF4 stimulated productive infection approximately 3-fold when DEX was added to cultures, which was significantly different than the effect observed with KLF4 alone. In summary, productive infection was stimulated the most by Sp3, KLF4, GR, and DEX treatment. These studies also implied that GR, Sp3, and KLF4 have multiple effects on viral replication because viral plaque formation was increased when DEX was added to cultures.

## 4. Discussion

BoHV-1 regulatory genes, including bICP0 and bICP4, are not abundantly expressed during latency in calves, as judged by in situ hybridization studies [53] and immunohistochemistry studies [26,27,28]. Consequently, stress-induced cellular transcription factors are predicted to trigger viral gene expression during the escape from latency to reactivation from latency. Notably, VP16 and bICP0 are detected prior to bICP4 and bICP22 [26,27,28]. bICP0 is a promiscuous transactivator of IE, E, and L promoters [54,55] and inhibits interferon responses [56,57]; consequently, bICP0 stimulates viral replication [58]. These results imply that the bICP0 E promoter is active prior to the IEtu1 promoter. Our previous studies demonstrating that two pioneer factors, GR [59] and KLF4 [60,61], synergistically transactivate the bICP0 E promoter [36,40] are intriguing because it is generally accepted that viral regulatory proteins are not abundantly expressed during latency because the viral promoters exist as silent heterochromatin [62,63]. Interestingly, KLF4 is detected in more TG neurons during the early stages of DEX-induced reactivation from latency when compared to latency and uninfected calves [30], indicating KLF4 is a stress-induced transcription factor.

In addition to being a pioneer transcription factor, KLF4 is one of four factors required to reprogram differentiated fibroblasts into inducible pluripotent cells [64]. Hence, KLF4 has many functions that regulate important cellular processes. For example, KLF4 promotes cell survival by impairing p53-dependent apoptosis, as reviewed in [65,66]. Surprisingly, other studies concluded KLF4 has pro-apoptotic functions [66,67], suggesting it has context-specific functions with respect to cell survival. KLF4 is also a tumor suppressor or an oncogene acting in a tissue dependent manner by regulating the expression of specific genes in distinct cell types and exhibiting pro- or anti-apoptotic functions. Unfortunately, the effect KLF4 plays in sensory neurons has not been studied. Sp1 [68,69] or Sp3 [70] over-expression can also induce apoptosis. Regardless of whether KLF4 expression in TG neurons during reactivation from latency stimulates or represses apoptosis, the ability of KLF4 to activate viral gene expression would occur. Since KLF4 clearly transactivates the bICP0 E promoter more efficiently than Sp1 or Sp3, we predict KLF4 is more important during reactivation from latency.

For the transactivation studies described in Figure 2, Figure 3, Figure 4 and Figure 5, DEX did not generally stimulate transactivation when GR was included in the transfection mixture. In certain cases, DEX reduced transactivation. This observation is consistent with ligand-independent GR activation [71]. In general, ligand-independent GR occurs when GR is phosphorylated by certain protein kinases. Consequently, phosphorylated GR is released from the heat shock complex in the cytoplasm and GR localizes to the nucleus. The ligand-independent GR-mediated transactivation of cis-regulatory modules is crucial for activating the HSV-1 ICP0 [72,73], ICP4 [74], and ICP27 [75] promoters. Ligand-independent GR phosphorylation is known to facilitate specific cellular stress signaling pathways [76,77]. Notably, ligand-independent GR phosphorylation occurs following UV exposure [78], which increases the expression of specific enzymes regulated by GR [78]. β_2_-adrenergic receptor agonists also induces ligand-independent GR activation [79]. Finally, epinephrine, a neurotransmitter released following stressful stimuli [79], activates GR [80,81].

Sp3 and GR had a cooperative effect on EP-638 promoter activity. Although GR and Sp1 had a modest cooperative effect on EP-638 promoter activity, neither Sp1 nor Sp3 cooperated with KLF4 to stimulate promoter activity. We expected Sp1 and Sp3 would yield the same results because they are expressed in most cells and are referred to as “basal” transcription factors, as reviewed in [31,82]. However, previous studies have demonstrated that the increased number of TG neurons in C57BL/6 mice express Sp3 during the early stages of DEX-induced explant-induced HSV-1 reactivation form latency. Conversely, Sp1+ TG neurons were only increased in female but not male C57Bl/6 mice during explant-induced reactivation from latency [35]. These two proteins also exhibit distinct intranuclear organizations [83]. Finally, Sp1 mediates early hematopoietic morphogenesis in mice [84], whereas Sp3 expression [85] promotes post-natal survival and tooth development. Electrophoretic mobility shift assays and chromatin immunoprecipitation assays revealed that activated GR and Sp3 bind the Cyclooxygenase 1 (COX-1) promoter, confirming these interactions increase COX-1 expression in cardiomyocytes [86]. Although Sp1 and Sp3 have many similarities, there are important differences in these Sp family members.

Recent studies revealed GR stably interacts with Sp1 [33] but not KLF4 (El-Mayet, unpublished studies). Studies demonstrating GR stably interacts with Sp3 were not identified in the literature, and we have not tested whether these transcription factors interact. It is also not clear whether Sp1 or Sp3 stably interact with KLF4. It is possible that either Sp1 or Sp3 indirectly interact with GR or KLF4 via interactions with other proteins. For example, GR indirectly interacts with basal transcription machinery via “tethering”, reviewed in [24]. Studies designed to resolve whether GR, Sp1, Sp3, and KLF4 directly interact with each or via a tethering related mechanism is being examined.

Three overlapping Sp1 binding sites (GGGCGG) and KLF4-like motifs were important for GR-and KLF4-mediated transactivation [36]. For this study, the C-rich Sp1 binding sites (CCGCCCC) and CACCCC motifs were not analyzed. These studies revealed that the CACCC binding site adjacent to the TATA box and five C-rich Sp1 consensus binding sites were also important. For example, the CACCCC motif that is only 10 nucleotides from the 5′ terminus of the bICP0 E promoter was crucial for the KLF4-mediated transactivation of the bICP0 E promoter. In addition to interacting with consensus Sp1 binding sites (GGGCGG) in key HSV-1 promoter sequences [74], KLF4 can also specifically bind CACCC motifs [87,88]. A previous study revealed GR and a CACCC-box binding factor cooperatively activate transcription [89,90]. In addition to KLF family members that interact with CACCC binding sites, Sp family members, the Wilms tumor gene, Vascular Endothelial Zinc Finger 1 (VEZF1), and Splat-Like Transcription Factor 4—also interact with CACCC motifs, as reviewed in [90].

## 5. Conclusions

These studies have demonstrated that GR and Sp3, but not KLF4, cooperatively stimulate EP-638 promoter activity and productive infection. Furthermore, Sp3 was more efficient than Sp1 at activating the bICP0 E promoter. These findings suggest there is a complex regulatory network in which GR, KLF4, and Sp family members activate BoHV-1 gene expression and productive infection. We have previously identified an antibiotic, Mithramycin A, that preferentially interacts with GC-rich DNA and impairs transcription [91]. Notably, treatment with Mithramycin A significantly reduces BoHV-1 [33] and HSV-1 replication in cultured cells [35], confirming that the transcription factors that bind GC-rich sequences, including KLF and Sp family members, are crucial for activating viral gene expression, viral replication, and reactivation from latency.

## Figures and Tables

**Figure 2 viruses-17-00229-f002:**
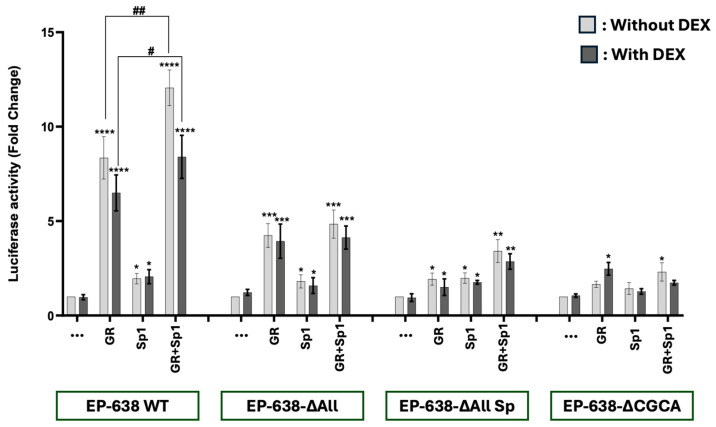
GR and Sp1 cooperatively transactivate bICP0 E promoter activity. Neuro-2A cells were co-transfected with the designated EP-638 promoter constructs (0.5 µg DNA), plasmids that express GR and/or Sp1 proteins (1.0 μg DNA of each). The designated cultures were treated with 2% stripped FBS, and DEX (10 μM) was added to cultures. At 48 h after transfection, cells were harvested and protein lysate subjected to dual-luciferase assay. Promoter activity levels in the sample containing wt EP-638 co-transfected with only an empty vector and only treated with DMSO (no hormone) were normalized to a value of 1, and fold activation for other samples presented. The results are the average of 3 independent experiments and error bars denote the standard error. Two-way ANOVA test was used for analyzing the results. Asterisks denote significant differences compared to the respective construct alone (*, *p* < 0.05; **, *p* < 0.01; ***, *p* < 0.001; and ****, *p* < 0.0001). A hash (#) denotes the significant differences between indicated groups (#, *p* < 0.05 and ##, *p* < 0.01) in the transactivation of wt EP-638 co-transfected with GR and Sp1 relative to Neuro-2A cells transfected with mutant EP-638 constructs.

**Figure 3 viruses-17-00229-f003:**
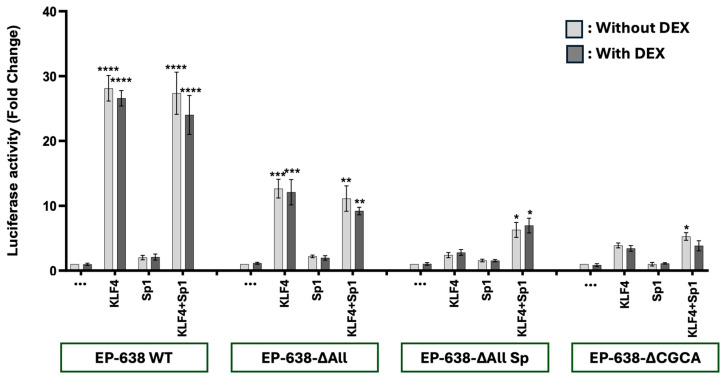
KLF4 and Sp1 do not cooperatively transactivate bICP0 E promoter activity. Neuro-2A cells were transfected with the designated EP-638 promoter constructs (0.5 μg DNA), plasmids that express KLF4 and/or Sp1 proteins (1.0 μg DNA of each). The designated cultures were treated with 2% stripped FBS, and DEX (10 μM) was added to cultures. At 48 h after transfection, cells were harvested and the protein lysate was subjected to dual-luciferase assay. Promoter activity levels in the sample containing wt EP-638 co-transfected with only an empty vector and only treated with DMSO (no hormone) were normalized to a value of 1, with fold activation for other samples presented. The results are the average of 3 independent experiments and error bars denote the standard error. Two-way ANOVA test was used for analyzing the results. An asterisk denotes significant differences compared to the respective construct alone *, *p* < 0.05; **, *p* < 0.01; ***, *p* < 0.001; and ****, *p* < 0.0001) in the transactivation of wt EP-638 co-transfected with KLF4 and Sp1 relative to Neuro-2A cells transfected with mutant EP-638 constructs.

**Figure 4 viruses-17-00229-f004:**
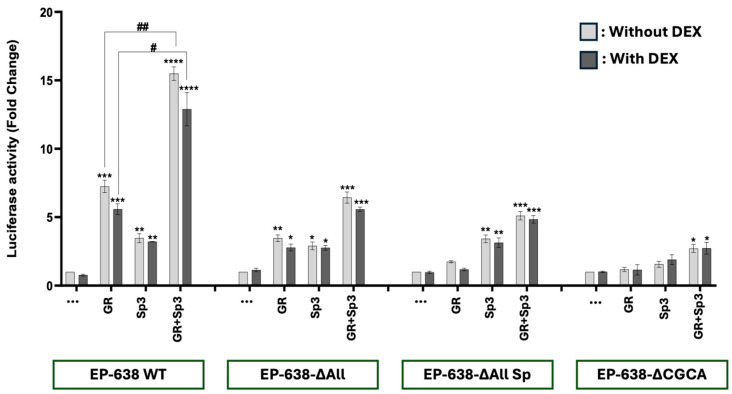
GR and Sp3 cooperatively transactivate the bICP0 E promoter. Neuro-2A cells were transfected with the designated EP-638 promoter constructs (0.5 μg DNA), plasmids that express GR and/or Sp3 proteins (1.0 μg DNA of each). Designated cultures were treated with 2% stripped FBS, and DEX (10 μM) was added to cultures. At 48 h after transfection, cells were harvested and protein lysate subjected to dual-luciferase assay. Promoter activity levels in the sample containing WT EP-638 co-transfected with only an empty vector and only treated with DMSO (no hormone) were normalized to a value of 1, with fold activation for other samples presented. The results are the average of 3 independent experiments and error bars denote the standard error. Two-way ANOVA test was used for analyzing the results. An asterisk denotes significant differences compared to respective construct alone (*, *p* < 0.05; **, *p* < 0.01; ***, *p* < 0.001; and ****, *p* < 0.0001). A hash (#) denotes significant differences between indicated groups (#, *p* < 0.05 and ##, *p* < 0.01) in the transactivation of wt EP-638 co-transfected with GR and Sp3 relative to Neuro-2A cells transfected with mutant EP-638 constructs.

**Figure 5 viruses-17-00229-f005:**
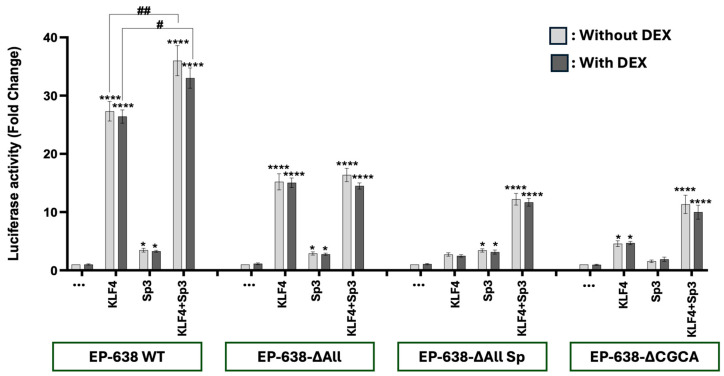
Sp3 increased the KLF4-mediated transactivation of the wt bICP0 E promoter. Neuro-2A cells were transfected with the designated EP-638 promoter constructs (0.5 μg DNA), KLF4 and/or Sp3 (1 μg DNA of each). The designated cultures were treated with 2% stripped FBS, and DEX (10 μM) was added to cultures. At 48 h after transfection, cells were harvested and protein lysate subjected to dual-luciferase assay. Promoter activity levels in the sample containing WT EP-638 co-transfected with only an empty vector and only treated with DMSO (no hormone) were normalized to a value of 1, with fold activation for other samples presented. The results are the average of 3 independent experiments and error bars denote the standard error. Two-way ANOVA test was used for analyzing the results. An asterisk denotes significant differences compared to the respective construct alone (*, *p* < 0.05; and ****, *p* < 0.0001). A hash denotes significant differences between indicated groups (#, *p* < 0.05 and ##, *p* < 0.01) in the transactivation of wt EP-638, co-transfected with a KLF4 and Sp3 relative to Neuro-2A cells transfected with mutant EP-638 constructs.

**Figure 6 viruses-17-00229-f006:**
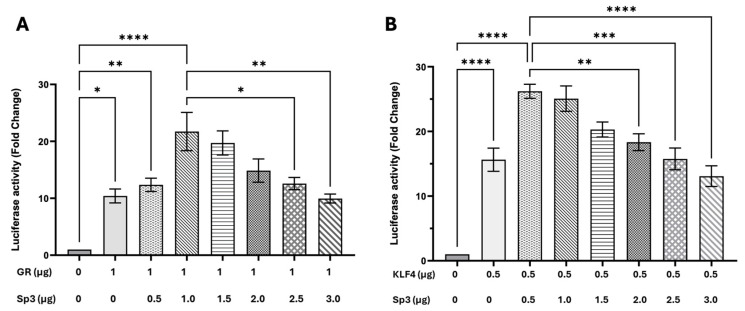
Analysis of cooperative interactions between GR and Sp3 or KLF4 and Sp3 of EP-638 promoter activity. (**A**) Neuro-2A cells were transfected with the EP-638 promoter construct (0.5 μg DNA), GR (1 μg DNA) and increasing concentrations of Sp3, as denoted. (**B**) Neuro-2A cells were transfected with the EP-638 promoter construct (0.5 μg DNA), KLF4 (1 μg DNA), and increasing concentrations of Sp3, as denoted. Promoter activity levels in the sample containing wt EP-638 co-transfected with only an empty vector were normalized to a value of 1, with fold activation for other samples presented. The results are the average of 3 independent experiments and error bars denote the standard error. Two-way ANOVA test was used for analyzing the results. An asterisk denotes significant differences compared to the respective construct alone (*, *p* < 0.05; **, *p* < 0.01; ***, *p* < 0.001; and ****, *p* < 0.0001) or transactivation of wt EP-638 co-transfected with GR and Sp3 in Neuro-2A cells.

**Figure 7 viruses-17-00229-f007:**
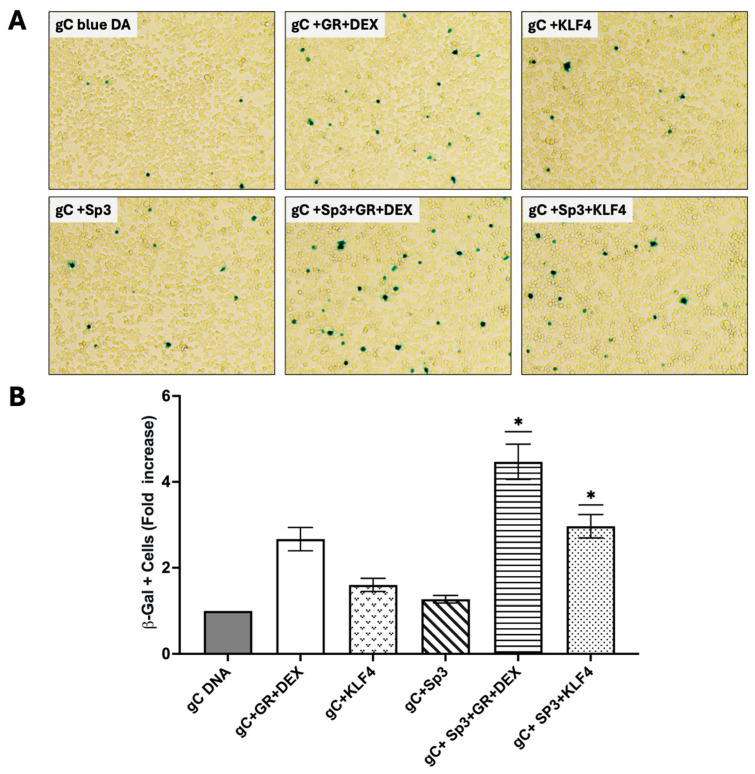
GR, Sp3, and/or KLF4 stimulate productive infection. (**A**) Neuro-2A cells were transfected with 1.0 μg BoHV-1 gCblue DNA and, where indicated, a plasmid expressing mouse GR protein (1.0 μg), Sp3 (1.0 μg), and/or KLF4 (1.0 μg). To maintain the same amount of DNA in each sample, an empty vector was included in the samples. After transfection, 2% stripped FBS was added to the medium. Designated cultures were then treated with water-soluble DEX (10 μM; Sigma) 24 h post-transfection. At 48 h after transfection, the number of β-Gal^+^ cells were counted. The value for the control (gCblue virus co-transfected with empty vector and then treated with PBS after transfection) was set to 1. (**B**) The results from DEX-treated cultures were compared to those from the control. The results are the average of 3 independent experiments and error bars denote the standard error. Two-way ANOVA test was used for analyzing the results. An asterisk denotes significant differences compared to the respective construct alone (*, *p* < 0.05).

## Data Availability

The original contributions presented in this study are included in the article. Further inquiries can be directed to the corresponding author.

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
