# Peer review of "Glucocorticoid Receptor (GR) and Specificity Protein 1 (Sp1) or Sp3 Transactivate the Bovine Alphaherpesvirus 1 (BoHV-1)-Infected Cell Protein 0 Early Promoter"

_viruses, 2025, doi:10.3390/v17020229_

Round 1
Reviewer 1 Report
Comments and Suggestions for Authors
In this manuscript, the authors have examined activation of bovine alphaherpesvirus 1 BoHV-1) infected cell protein 0 (bICP0) early promoter (EP-638) by combinations of glucocorticoid receptor (GR), Kruppel-like factor 4 (KLF4) and specificity (Sp) proteins 1 and 3. Their main objective is to determine if Sp1 and Sp3 cooperate with GR to transactivate bICP0 E promoter and stimulate virus replication. They do demonstrate that SP3, along with GR and KLF4, cooperatively stimulated EP-638 and induced virus replication. BoHV-1 is an important pathogen associated with the cattle industry because it causes life-threatening diseases. They have achieved their objectives and the data are convincing. However, addressing the following points will help the readers.
Figure 1 and the legend is very confusing and needs to be rewritten for clarity.
”KLF4+HR” is not explained in the legend.
The differences between KLF’s indicated by different colored circles are not clear.
The difference between -É…All and -É…AllSp includes the missing the “CACCC” sequence close to the TATA box and SP1 site close to position 328, which is not mentioned.
-É…CGCA mutant needs to be described better with details.
Lines 227-30: In addition to the absence of CACC-rich motif adjacent to the TATA box, SP1 site close to position 32 is also missing.
Lines 272-4: Based on Fig 4, GR-mediated transactivation seems to lower than SP3 using EP638- É…All.
Line 330: Need to explain why they excluded SP1 even though it cooperatively transactivated with GR. They do address it in lines 380-383. It would be good to reiterate it before describing the experiments in Fig 7.
Minor points:
Line 323: “#” indicated in the legend is absent in the Fig.
Line 345: should be Figure 7A and B.
Line 176: should read, “previous study which tested whether…..”
Author Response
With respect to the comments of Reviewer 1, the following changes have been made in the revised manuscript.
Concern #1: Figure 1 and the legend is very confusing and needs to be rewritten for clarity.
Response: I agree Figure 1 is complicated. The figure legend has been carefully revised and it now summarizes the information presented in Figure 1.
Concern #2: KLF4+HR” is not explained in the legend.
Response: It should have been denoted as KLF4 and GR and this described in the legend of Figure 1 (lines -96-97).
Concern #3: The differences between KLF’s indicated by different colored circles are not clear.
Response: There are several KLF4 binding sites that have been identified. These sequences depend on whether they are chromatinized or what transcriptional coactivators can bind to adjacent sequences. These point were added to the legend of Figure 1 (lines 100-102). Distinct colors that are easily discerned denote the important transcription binding sites in Panel C.
Concern #4: The difference between -É…All and -É…AllSp includes the missing the “CACCC” sequence close to the TATA box and SP1 site close to position 328, which is not mentioned.
Response: This information is included in the revised manuscript (lines 198-200).
Concern #5: -É…CGCA mutant needs to be described better with details.
Response: This information is included in the revised manuscript (lines 206-208 and 215-218).
Concern #6: Lines 227-30: In addition to the absence of CACC-rich motif adjacent to the TATA box, SP1 site close to position 32 is also missing.
Response: I am sorry but lines 227-30 of the original manuscript did not discuss the CACC-rich motif nor a Sp1 site close to position 32. With that said, we have carefully explained these mutants in the revised manuscript.
Concern #7: Lines 272-4: Based on Fig 4, GR-mediated transactivation seems to lower than SP3 using EP638- É…All.
Response: This is true and is now stated in the revised manuscript (lines 279-286).
Concern #8: Line 330: Need to explain why they excluded SP1 even though it cooperatively transactivated with GR. They do address it in lines 380-383. It would be good to reiterate it before describing the experiments in Fig 7.
Response: We pointed out that a Sp1 siRNA inhibited BoHV-1 replication and that is why we focused on the effect Sp3, GR, and KLF4 had on viral replication (lines 361-365).
Concern #9: Line 323: “#” indicated in the legend is absent in the Fig.
Response: Each Figure legend in the revised manuscript describes what the asterisks and # mean.
Concern #10: Line 345: should be Figure 7A and B.
Response: This was corrected (lines 374- 376).
Concern #11: Line 176: should read, “previous study which tested whether…..”
Response: This sentence was modified (lines 180-181).

Reviewer 2 Report
Comments and Suggestions for Authors
Bovine alphaherpesvirus 1 (BoHV-1) infection reduces host defenses, affecting mucosal surfaces, and increases susceptibility to bacterial pneumonia, contributing to bovine respiratory disease (BRD), which is a leading cause of morbidity and mortality in cattle. The virus utilizes host receptors for cell entry, and stress or co-infections with bacteria like Mannheimia haemolytica can exacerbate bacterial pneumonia. BoHV-1 infection can result in cell death, inflammation, and robust virus production, with its gene expression occurring in a gene cascade (immediate early (IE), early (E), and late (L)) that coordinate productive infection through important IE regulatory proteins such as bICP0 and bICP4.
BoHV-1 establishes latency in neurons of the trigeminal ganglia (TG), and stress conditions can elevate corticosteroid levels, triggering viral reactivation. The synthetic corticosteroid dexamethasone (DEX) activates the glucocorticoid receptor (GR), leading to the rapid expression of IE genes like bICP0 and bICP4. Interestingly, bICP0 is detected first, suggesting an earlier activation of the bICP0 E promoter compared to the IEtu1 promoter. GR with Krüppel-like transcription factors (KLFs) - a part of the Sp family of transcription factors - and other stress-induced factors synergistically transactivates viral promoters, facilitating productive infection during reactivation. In the current study, the authors examined the roles of host transcriptions factors Sp1, Sp3, GR, and KLF4 play to cooperatively transactivate the bICP0 E promoter and stimulate BoHV-1 productive infection in a murine neuronal cell line.
Experiments with the wildtype (WT) bICP0 E promoter construct revealed that Sp1 can transactivate the bICP0 E promoter only in cooperation with GR, but it does not synergize with KLF4, which alone robustly stimulates the promoter. In contrast, Sp3, in cooperation with GR, enhances bICP0 E promoter activity and also increases KLF4-mediated transactivation in an additive manner. Additionally, Sp3 stimulates productive infection when co-expressed with GR or KLF4. Notably, the EP-638 mutants exhibited overall reduced transactivation of the bICP0 E promoter by the tested factors, either individually or cooperatively, highlighting the importance of intact motifs for both individual and cooperative activity. The EP638-∆CGCA construct, which showed the lowest transactivation, revealed that the CACCC motifs near the TATA box are critical for promoter activity and exert a stronger influence than upstream factors binding sites. The authors data suggest that interactions between Sp1 and GR and Sp3 and GR or KLF4 may regulate bICP0 E expression during the early stages of reactivation. Experiments presented in this study were thorough, and their results were straightforward. Specific points regarding the manuscript as discussed below.
- The role of DEX in Figures 2-6. In all experiments investigating the transactivating activity of bICP0, DEX treatment either reduces activity or has no effect. Given that DEX stimulates GR, can the authors explain why DEX does not have a stimulatory effect in their reporter assays? Please explain. Also, for these figures, what does ## and/or **, ***, and **** mean in these figures? The meanings are not provided in the figure legends. Please include what these symbols represent (e.g., greater significance, p<0.01).
- Lines 205–206: Statements about significant effects should be accompanied by statistical significance values.
- Synergistic effects of transcription factors (Figures 2, 4, and 5). Do the authors know if GR and Sp1, GR and Sp3, and KLF4 and Sp3 interact with one another (dependent or independent of DNA binding) to provide a potential mechanism of their cooperative effects on transcription? Please comment about this in the Discussion.
- Lines 224–225: While the effect of Sp1 alone is negligible in Figure 2, it is still statistically significant, unlike its effect in Figure 3. Why is this?
- Lines 269–275: Statistical analyses should be provided in Figure 4 or the text to support the claim that there was no cooperative transactivation between GR and Sp3 for the EP638-∆All mutant compared to GR alone (as the graph suggests a near two-fold difference). Similarly, statements about the EP638-∆AllSp and EP638-∆CGCA mutants should be supported by statistical evidence.
Minor Points
- Figures 2 and 4: The horizontal bars indicating statistical comparisons above the bar graphs should be smaller to clearly show which bars below are being compared.
- Lines 223–224: There is a gap in the middle of the sentence that should be removed.
- Line 285: The reference to the EP638-∆AllSp mutant in the text should be corrected to EP638-∆All.
- Line 345: Replace "Figures 6A and B" with "Figures 7A and B" in the text.
Author Response
With respect to the comments of Reviewer #2, the following changes have been made in the revised manuscript.
Concern #1: The role of DEX in Figures 2-6. In all experiments investigating the transactivating activity of bICP0, DEX treatment either reduces activity or has no effect. Given that DEX stimulates GR, can the authors explain why DEX does not have a stimulatory effect in their reporter assays? Please explain.
Response: DEX did not stimulate GR-mediated transactivation and in certain instances it reduced transactivation. This has been a consistent effect on the bICP0 E promoter and certain HSV-1 promoters. These results are consistent with transactivation being mediated by “ligand-independent” GR activation. GR can be directly activated by phosphorylation of certain important serine or threonine residues by specific protein kinases. This is discussed in the revised manuscript (lines 427-439).
Concern #2: Also, for these figures, what does ## and/or **, ***, and **** mean in these figures? The meanings are not provided in the figure legends. Please include what these symbols represent (e.g., greater significance, p<0.01).
Response: This important information has been added to each Figure Legend of the revised manuscript.
Concern #3: Lines 205–206: Statements about significant effects should be accompanied by statistical significance values.
Response: Agreed. This information is included in the Figure Legends of the revised manuscript.
Concern #4: Synergistic effects of transcription factors (Figures 2, 4, and 5). Do the authors know if GR and Sp1, GR and Sp3, and KLF4 and Sp3 interact with one another (dependent or independent of DNA binding) to provide a potential mechanism of their cooperative effects on transcription? Please comment about this in the Discussion.
Response: Thank you for bringing this important point up. We previously demonstrated GR stably interacts with Sp1 but not KLF4. We have not tested whether GR stably interacts with Sp3, in part because we have not found a good Sp3 antibody that works for co-IP studies. Furthermore, I was unable to find a publication that states GR does or does not interact with Sp3. It is also possible there may be complexes that contain GR and Sp3. This information is included in the revised manuscript (lines 455-462).
Concern #5: Lines 224–225: While the effect of Sp1 alone is negligible in Figure 2, it is still statistically significant, unlike its effect in Figure 3. Why is this?
Response: For Figure 2, GR transactivated the EP-638 wild-type promoter construct less than 10-fold: hence, a 2-fold increased promoter activity by Sp1 was significantly higher than the control. Since, KLF4 transactivated the wild-type EP-638 ~30-fold, the 2-fold increase by Sp1 was not significantly different.
Concern #6: Lines 269–275: Statistical analyses should be provided in Figure 4 or the text to support the claim that there was no cooperative transactivation between GR and Sp3 for the EP638-∆All mutant compared to GR alone (as the graph suggests a near two-fold difference). Similarly, statements about the EP638-∆AllSp and EP638-∆CGCA mutants should be supported by statistical evidence.
Response: This was included in the revised manuscript (lines 279-286 and 295-301 in the legend of Figure 4).
Concern #7: Figures 2 and 4: The horizontal bars indicating statistical comparisons above the bar graphs should be smaller to clearly show which bars below are being compared.
Response: This was corrected in Figures 2 and 4 of the revised manuscript.
Concern #8: Lines 223–224: There is a gap in the middle of the sentence that should be removed.
Response: the gap was removed in the sentence (lines 242-244)
Concern #9: Line 285: The reference to the EP638-∆AllSp mutant in the text should be corrected to EP638-∆All.
Response: Actually, there are two constructs; EP638-∆All and EP638-∆AllSp. We have carefully checked the manuscript and these constructs are correctly denoted in the revised manuscript
Concern #10: Line 345: Replace "Figures 6A and B" with "Figures 7A and B" in the text.
Response: This was corrected (lines 374- 376).

Reviewer 3 Report
Comments and Suggestions for Authors
The authors provide interesting new information on the role of GR, Sp1, Sp3, and KLF4 on bovine ICP0 promoter activities in vitro. The authors have shown that these transcription factors enhance viral replication and gene expression. In addition, they have shown that Sp3 play more significant role than Sp1. Overall, the manuscript provides new and essential information, and the information are highly significant with regards to control of BoHV-1 infection and pathogenesis. This information can be used to study many new aspects of BoHV-1 infection-replication including the potential of using their approach to block viral replication. The manuscript is very professionally written, and the data support the conclusion. Listed below are some points for consideration by the authors:
1. Can the authors discuss their finding in relation to HSV-1?
2. Since vaccines are available against BoHV-1 infection, can the authors discuss the importance of their finding with regards to improvement of the available vaccines.
3. In the discussion, the authors should discuss and speculate the clinical application of their important findings.
4. KLFs are belong to SP1 family and the authors showed KLF4 is more important than Sp1. Can the authors discuss the relationship of Sp1 and KLF4 in the context of their study. Also, Sp1 family are involved in apoptosis. Is the apoptosis function of Sp1 family associated with KLF4?
5. Please use two-sided error bars for all figures.
Author Response
With respect to the comments of Reviewer #3, the following changes have been made in the revised manuscript.
Concern #1: Can the authors discuss their finding in relation to HSV-1?
Response: See lines 359-360, 433-434, 444-446, 469-471, and 482-486.
Concern #2: Since vaccines are available against BoHV-1 infection, can the authors discuss the importance of their finding with regards to improvement of the available vaccines.
Response: We have generated a mutant BoHV-1 virus where the TATA box of the bICP0 E promoter was mutated. This virus grows poorly in calves, does not cause disease, and reactivates poorly. However, infection with the mutant leads to high levels of antibody production. Since these studies are in progress, it is a bit premature to discuss this.
Concern #3: In the discussion, the authors should discuss and speculate the clinical application of their important findings.
Response: See 481-486.
Concern #4: KLFs belong to SP1 family and the authors showed KLF4 is more important than Sp1. Can the authors discuss the relationship of Sp1 and KLF4 in the context of their study. Also, Sp1 family are involved in apoptosis. Is the apoptosis function of Sp1 family associated with KLF4?
Response: KLF4 can inhibit or promoter apoptosis in a cell-type dependent manner. To date, it is not clear whether KLF4 has pro- or anti-apoptotic functions in sensory neurons. These concepts were added to the discussion (lines 413-426).
Concern #5: Please use two-sided error bars for all figures.
Response: All figures now contain 2-sided error bars.

Round 2
Reviewer 2 Report
Comments and Suggestions for Authors
The authors have appropriately addressed the reviewers' concerns and have strengthen their manuscript in their edited resubmission.